# Synucleins: New Data on Misfolding, Aggregation and Role in Diseases

**DOI:** 10.3390/biomedicines10123241

**Published:** 2022-12-13

**Authors:** Andrei Surguchov, Alexei Surguchev

**Affiliations:** 1Department of Neurology, Kansas University Medical Center, Kansas City, KS 66160, USA; 2Section of Otolaryngology, Department of Surgery, Yale School of Medicine, Yale University, New Haven, CT 06520, USA

**Keywords:** α-, β- and γ-synuclein, Parkinson’s disease, epigenetic, protein aggregation, SNARE-complex, protein trafficking, phytochemicals, synucleinopathies, methylation, histones

## Abstract

The synucleins are a family of natively unfolded (or intrinsically unstructured) proteins consisting of α-, β-, and γ-synuclein involved in neurodegenerative diseases and cancer. The current number of publications on synucleins has exceeded 16.000. They remain the subject of constant interest for over 35 years. Two reasons explain this unchanging attention: synuclein’s association with several severe human diseases and the lack of understanding of the functional roles under normal physiological conditions. We analyzed recent publications to look at the main trends and developments in synuclein research and discuss possible future directions. Traditional areas of peak research interest which still remain high among last year’s publications are comparative studies of structural features as well as functional research on of three members of the synuclein family. Another popular research topic in the area is a mechanism of α-synuclein accumulation, aggregation, and fibrillation. Exciting fast-growing area of recent research is α-synuclein and epigenetics. We do not present here a broad and comprehensive review of all directions of studies but summarize only the most significant recent findings relevant to these topics and outline potential future directions.

## 1. Introduction

Synucleins are a family of small, prone to aggregate intrinsically disordered proteins (IDP) implicated in neurodegenerative diseases and cancer. Since synucleins are involved in severe human diseases, understandably most research is directed at unveiling their role in pathology. As a result, we know much more about their contribution to pathological mechanisms than their normal functions, which are still not fully understood. 

The first synuclein was identified using antiserum against purified cholinergic synaptic vesicles as a 143 amino acid presynaptic protein in the electric organ of *Torpedo californica* [1]. Later, two additional isoforms belonging to the synuclein family were identified [2]. Three evolutionary conserved members of this family are highly expressed in the vertebrate nervous system and have been found in all vertebrates [3]. Importantly, no counterparts of synucleins were identified in invertebrates, indicating that they are vertebrate-specific proteins. Further analysis demonstrates that the number of synuclein members may differ among vertebrates. While three genes encoding α-, β-, and γ-synuclein are present in mammals and birds, a variable number of synuclein genes and corresponding proteins are expressed in fish, depending on the species. For example, four synuclein genes are identified in fugu, encoding for α, β, and two γ (g1 and g2) isoforms, but three genes are detected in zebrafish [4,5].

IDPs constitute about one-third of the human proteome [6] pointing to the importance of their structural organization. The absence of a defined structure confers members of the synuclein family conformational flexibility, allowing them to participate in dynamic and transient molecular interactions.

## 2. Three Members of The Synuclein Family

The overwhelming majority of publications on synucleins are dedicated to α-synuclein due to its involvement in Parkinson’s disease (PD) and other synucleinopathies. Yet, two other proteins β- and γ-synucleins belonging to the same family are localized predominantly in neuronal terminals and implicated in the long-term modulation and preservation of nerve terminal function and dopamine homeostasis [7]. As a result, they deserve more attention to better understand the functionality of nervous system and role in carcinogenesis.

### 2.1. Common Structure of Members of the Synuclein Family

As shown in Figure 1 synucleins have a substantial degree of amino acid sequence and overall structural similarities (Figure 1).

The members of the synuclein family mostly share a structurally similar N-terminal amino-acid sequence; α-synuclein supposedly forms a single or two distinct α-helices in this region. This transition is induced when α-synuclein interacts with the membranes via lipid bilayers. A C-terminal region contains negatively charged residues and prolines. The C-terminal domain differs in amino acid sequence among synucleins and regulates their solubility depending on its length and charge. A stretch of amino acids located between positions 61–95 in α-synuclein is called NAC (Figure 1), and a stretch of amino acids 71–82 within the α-synuclein NAC region is critical for β-sheet-rich aggregation in vitro [8,9]. This amino acid sequence is VTGVTAVAQKTV in α-synuclein, FSGAGNIAAATG in β-synuclein, and VSSVNTVATKTV in γ-synuclein. Amino acids belonging to the 71–82 stretch are necessary and sufficient for α-synuclein fibrillization [9].

Furthermore, the substitutions of amino acids (E35K + E46K + E61K = ‘3K’) are render all synucleins considerably more toxic than their wild-type counterparts [9]. These substitutions increase α-synuclein-membrane interactions pointing to a mechanism of synuclein toxicity associated with membrane binding. Binding to lipids converts the N-terminal end to an α-helix. The N-terminal helicity negatively correlates with aggregation potential, while the C-termini differ among synucleins regulating their solubility.

Newberry et al. [13] used mutational scanning to reveal the association between its structure, activity and toxicity. They analyzed a library of protein missense variants for relative toxicity in competitive selection in yeast cells. In the headgroup region, mutations of Lys residues, which are expected to interact with acidic residues in anionic lipids, strongly decreased toxicity. Membrane binding of α-synuclein and the formation of a helix in the membrane-bound state of α-synuclein determine the level of α-synuclein toxicity. These results point to the importance of conserved lysine residues in membrane-bound α-synuclein for its toxicity. Such an approach allowed the generation of a high-resolution model for the structure and dynamics of the conformational state of α-synuclein that slows yeast growth [13].

### 2.2. Synuclein’s Cellular Functions and Role in Pathology

α-Synuclein is an essential regulator of synaptic vesicle pool and trafficking, dopamine neuro-transmission, and other mechanisms involved in synaptic plasticity [14,15]. α-Synuclein performs these functions by assisting in the creation of soluble N-ethylmaleimide-sensitive-factor attachment protein receptor (SNARE)-complex [15,16,17,18]. α-Synuclein participates in chaperoning SNARE complex assembly through its interaction with vesicle-associated membrane protein 2 (VAMP2)/synaptobrevin-2 [14].

Contrary to α-synuclein, our knowledge about the role of β-synuclein and γ-synuclein in cellular functions and pathology is somewhat limited. Recent findings about β-synuclein point to its role as an important biomarker for the early stages of Alzheimer’s disease (AD) [19]. The concentration of β-synuclein gradually increases in the cerebrospinal fluid beginning from the preclinical AD phase. It may be considered a promising biomarker of synaptic damage in this disease. β-Synuclein may be used as a CSF biomarker for synaptic damage in AD; its level is elevated in both dementia and pre-dementia stages of AD [19,20]. Importantly, higher CSF α-synuclein levels are reported in pre-AD subjects but not in MCI-AD and dementia AD [19], pointing to its specificity as a biomarker.

Other significant findings concerning β-synuclein and γ-synuclein have been described recently by Carnazza et al. [3]. These researchers have demonstrated that both β-synuclein and γ-synuclein have a lower affinity toward synaptic vesicles than α-synuclein. β-Synuclein and, to a lesser extent, γ-synuclein target to the synapse, despite a dramatically reduced ability to associate with membranes compared with α-synuclein. Formation of oligomers of β-synuclein or γ-synuclein with α-synuclein causes a reduction of synaptic vesicle binding of α-synuclein. These results indicate that β-synuclein and γ-synuclein are modulators of α-synuclein binding to synaptic vesicles. Therefore, they may reduce α-synuclein’s physiological activity at the neuronal synapse. Another conclusion from these results is that members of the synuclein family can regulate each other’s ability to bind to synaptic vesicles [3]. Inactivation of γ-synuclein affects psycho-emotional status and cognitive abilities. Moreover, in aging mice lacking γ-synuclein the dopaminergic dysfunction and altered working memory performance was demonstrated [21].

After oxidation on methionine and tyrosine located in neighboring positions, Met^38^ and Tyr^39^ γ-synuclein forms annular oligomers and can initiate α-synuclein aggregation [22]. 

Examination of synucleins in a non-human primate model of PD showed that the expression of all three members of the family was increased in the substantia nigra after treatment with prodrug to the neurotoxin-1-methyl-4-phenyl-1,2,3,6-tetrahydropyridine (MPTP). This increase correlates positively with the loss of cells and motor score [7]. The expression of the three types of synucleins was also increased in the dorsal raphe nucleus, but only α- and γ-synucleins are associated with the loss of cells and cell loss. The authors conclude that MPTP treatment mainly affects the expression level of α-synuclein and γ-synucleins and, to a lesser extent, β-synuclein levels and suggest that the three members of the synuclein family participate in the regulation of monoaminergic transmission [7]. 

In another study of the functional role of three members of the family, Ninkina et al. [23] found that the monoamine transporter 2–dependent dopamine uptake by synaptic vesicles of mice with β-synuclein knockout is considerably diminished. Introduction of β-synuclein recovers uptake by triple α/β/γ-synuclein–deficient striatal vesicles; however, α-synuclein or γ-synuclein do not cause such recovery. Interestingly, only β-synuclein potentiates VMAT-2–dependent uptake of dopamine and structurally similar molecules by synaptic vesicles. The authors propose that the increased presence of β-synuclein-triggered complexes at the synaptic vesicles explains the decreased sensitivity to MPTP toxicity of SNpc dopaminergic neurons in mice lacking α-synuclein and or γ-synuclein.

An association of α-synuclein and γ-synuclein with autism spectrum disorder (ASD) has recently been described. A significant reduction of plasma levels of α-synuclein and elevation of plasma γ-synuclein in children with ASD was found [24]. Plasma levels of both synucleins are significantly related to the severity of ASD and might serve as an indicator of the rigorousness of the disorder. In particular, the plasma level of γ-synuclein is lower in ASD patients than in controls, which the authors explain by disruption of the synaptic functions [24].

One of the possible mechanisms of γ-synuclein involvement in diseases may be associated with its role in the control of metabolic functions in fat cells. Rodríguez-Barrueco et al. [25] recently revealed details of the regulatory mechanism of fat mass expansion modulated by the micro-RNA [miR-424(322)/503] through γ-synuclein. This miRNA plays a crucial role in adipose tissue regulating impaired adipogenesis and fat mass via the control of γ-synuclein [25]. The authors identified and validated the direct influence of miR-424(322)/503 on the level of γ-synuclein expression and consider it a target gene in mediating the metabolic functions of adipocytes.

## 3. α-Synuclein Misfolding, Aggregation, and Fibrillation

Intracellular accumulation of prone-to-aggregate proteins associated with the formation of amyloid-like fibrils, is a common neuropathological feature of several neurodegenerative diseases. The “unfolding” of intrinsically disordered proteins is similar to the misfolding of globular proteins [26,27].

Under physiological conditions, α-synuclein in the cytoplasm exists in a dynamic equilibrium between soluble monomers and a variety of oligomers, including helically folded tetramers (Figure 2) [28,29,30].

Binding to lipid membranes is a key mediator of oligomerization and aggregation of α-synuclein. α-Synuclein-containing protein inclusions, aggregates, and fibrils are found in neuropathological inclusions in neurons and glia of patients with synucleinopathies [28]. Various mutations and genomic multiplications, including duplication and triplication of the α-synuclein gene cause familial PD with extramotor features [31].

It is generally supposed that α-synuclein may cause diseases via a toxic gain-of-function intrinsic for the normal protein when it surpasses a certain accumulation level. In agreement with this concept, α-synuclein knockout mice, in contrast to transgenic overexpressors, display no obvious neuropathological phenotype [14,32]. More detailed investigations reveal deficits in such null mice’ dopaminergic system. The role of loss of α-synuclein functions in pathology due to the sequestration of physiological α-synuclein in deposits and inclusions cannot be completely excluded. It may damage some of the synaptic function of α-synuclein when its concentration falls below some critical level.

### 3.1. Post-Translational Modifications (PTMs) of α-Synuclein

Various PTMs regulate α-synuclein susceptibility to aggregation by changing its conformation. PTMs such as phosphorylation, ubiquitination, acetylation, nitration, SUMOylation, etc., modify α-synuclein physicochemical properties and affect the aggregation process in synucleinopathies [11,12,29,31,32,33,34]. α-Synuclein is acetylated constitutively at its N-terminal (Figure 1). As Bell et al. recently demonstrated [11,12] this PTM changes the charge and spatial structure of α-synuclein and affects its aggregation and interaction with lipid membranes. Even subtle disturbances caused by PTMs as well as mutations, can lead to remarkable alterations in the aggregation performance of this protein [11,12].

In addition to PTMs, many other factors are able to induce aggregation-prone conformations of α-synuclein such as focal changes in pH or Ca^2+^ concentration. For instance, Ca^2+^ binding to α-synuclein C-terminus induces N-terminal unfolding and aggregation-prone conformations [35]. At low pH, the large net negative charge at the C-terminus is reduced, diminishing charge-charge intramolecular repulsion. This change causes conversion of α-synuclein to partially unfolded conformation. Proteolytic truncation of α-synuclein C-terminus also accelerates its aggregation and increases toxicity. The C-terminus determines α-synuclein cytotoxicity and aggregation by regulating binding to membranes and chaperones [36].

### 3.2. Approaches to Reduce α-Synuclein Toxicity

One approach to neutralizing α-synuclein toxicity, which offered a lot of promise, has been targeting the pathologically aggregated form of α-synuclein. Since aggregated protein plays an essential role in synucleinopathies’ pathogenesis, many attempts have been made to reduce α-synuclein accumulation and aggregation. One method has been based on the antibody specific to the aggregated form of α-synuclein [37,38]. Recently, the monoclonal antibody prasinezumab, directed at aggregated α-synuclein, has been studied as a potential treatment for PD (Pasadena trial) [39]. In a phase 2 double-blind trial, patients with earlyPD were treated by intravenous placebo or prasinezumab. The trial results were similar in the treatment groups and the placebo group indicating that prasinezumab therapy had no meaningful effect on global or imaging measures of PD progression as compared with placebo. Lang et al. [40] tested another monoclonal antibody—cinpanemab (SPARK trial) using a similar trial design with the same disappointing result indicating that both potential antibodies have no implications for current practice. 

However, these negative and discouraging results should not be taken wrongly as proof that targeting α-synuclein is an unsuccessful approach. The negative result may be explained by several reasons. For example, it may be due to the failure of antibodies to enter the brain in sufficient amounts for a noticeable therapeutic effect. There are reasons to believe that antibodies that are delivered intravenously do not usually pass through intact cell membranes, including those of the blood-brain barrier [37]. Another explanation of the absence of therapeutic efficacy of these antibodies may be inability of antibodies to traverse cell membranes and penetrate intraneuronal space or cross the surface of exosomes. Thus, α-synuclein may be just inaccessible to antibodies. Potential tactics to overcome limitations of antibody therapies for synucleinopathies may be based on enhancing blood–brain barrier penetration with technologies such as magnetic resonance-guided focused ultrasound [41]. An alternative approach is developing modified antibodies that will be able to penetrate cell membranes [42]. Therefore, despite the failure of these clinical trials, there is still a chance that targeting α-synuclein will open an approach for molecular therapy for PD [37]. In other words, these results of clinical trials should not dismiss the possibility that successful results may yet be attained with the same or analogous antibody in prodromal PD or in genetic forms of this disorder. Alternative tactics to affect aggregated α-synuclein may be also beneficial [38].

### 3.3. The Quaternary Structure of α-Synuclein Fibrils Modulates α-Synuclein Pathology

Frieg et al. [43] amplified α-synuclein fibrils from brain extracts of patients with pathologically confirmed PD and multiple system atrophy, determined their 3D structures by Cryo-EM, and evaluated their potential to seed α-synuclein related pathology in oligodendrocytes. In the first step, researchers fibrilized α-synuclein by protein misfolding cyclic amplification (PMCA) (see Section 3) using brain extracts of patients. This approach showed that α-synuclein aggregated by incubation with homogenates from the brain tissues of these patients determined the fibril structures. Then, the authors analyzed fibrils by Cryo-EM and other methods and characterized their seeding potential in mouse primary oligodendroglial cultures. This examination showed distinct 3D structures and quaternary arrangement of α-synuclein fibrils.

The data demonstrate that the quaternary structure of α-synuclein fibrils is a vital factor in the seeding of pathology. Investigation of α-synuclein fibrils by Cryo-EM has shown that the dominant fibril morphology is rod polymorph [44]. Furthermore, the amplified α-synuclein fibrils differed in their inter-protofilament interface and their ability to recruit endogenous α-synuclein depending on the origin of the patient’s homogenate. The fibrils were also different in their adopted helical arrangement and N-terminal region. Thus, the quaternary structure of α-synuclein fibrils modulated α-synuclein pathology inside recipient cells.

Interestingly α-synuclein monomers, oligomers and fibrils exert a differential effect on the folding and refolding of other proteins. Bagree et al. [45] examined their influence on the conformation, enzymatic activity and other properties of firefly luciferase. α-Synuclein monomers delayed the inactivation of luciferase under thermal stress conditions and enhanced the spontaneous refolding, whereas oligomers and fibrils adversely influenced luciferase activity and refolding. The oligomers suppressed spontaneous refolding, while fibrils caused a pronounced effect on the inactivation of native enzyme. Thus, various conformers of α-synuclein exert differential effect on structural modifications and misfolding of other proteins, regulating protein homeostasis.

## 4. Synuclein-Based Methods of Disease Diagnosis

Since α-synuclein misfolding and aggregation precede major clinical manifestations, detecting misfolded and aggregated α-synuclein would allow recognition of the disease at the earliest premotor phases. Further accrual and spreading of these pathological isoforms of α-synuclein usually correlate with clinical symptoms and progression of synucleinopathies. Recent findings point to a mechanism of template-mediated amplification of amyloid-like fibrils as a base of accumulation and intracellular propagation of fibril seeds by which pathological features spread along the neural circuits in the brain. Abnormal accumulation of α-synuclein [15,46], and in some cases of γ-synuclein [47,48], occurs in synucleinopathies [49]. Although relatively small α-synuclein oligomers are considered the most toxic species, it is not exactly known which α-synuclein assemblies possess the most toxic properties [26] and are the most relevant to human diseases. Accumulating data indicate that smaller-size aggregates cause a higher toxic response than filamentous aggregates (fibrils) [50]. The application of fluorophores for detecting various α-synuclein subtypes brings the hope that the initial steps of aggregation and the structure of intermediates will be better understood. A commercial aggregate-activated fluorophore Amytracker 630 (AT630) possesses photophysical properties that enable high-resolution (∼4 nm) imaging of α-synuclein and other aggregation-prone proteins. Using fluorophore AT630, the authors found that aggregates smaller than 450 ± 60 nm easily penetrated the plasma membrane. An important conclusion from this study is that aggregates in different synucleinopathies, i.e., PD and DLB have different potency in toxicity. Moreover, there was a linear relationship between α-synuclein aggregate size and cellular toxicity. This novel method allows for the measuring of toxic protein aggregates in biological environments quantitatively and therefore brings a better understanding of disease mechanisms under physiological conditions [50].

Sekiya et al. [51] used proximity ligation assay—a new technique to detect the distribution of α-synuclein oligomers, compare the results with immunohistochemical data and analyze the correlation between the presence of oligomers with clinical features. The results show that α-synuclein oligomers are more widespread than Lewy-related pathology (LRP, Lewy bodies, and Lewy neurites) and that α-synuclein oligomers in the hippocampus correlate with cognitive impairment. Moreover, there was discordance between the distribution of α-synuclein oligomers and LRP. In about one-half of patients, LRP were not found in the neocortex, but at least some of the α-synuclein oligomers were detected in the neocortex of all patients. More α-synuclein oligomers accumulated in the neuropil than in neurons in most brain regions. Furthermore, α-synuclein oligomers may be widespread early in the disease stage. These results may explain (at least partially) the recent failure of anti-α-synuclein therapy [52]. The unsuccessful attempts of this cure may be due to the widespread distribution of α-synuclein oligomers in earlier pathological stages of the disease. To be successful, the effective treatment of PD patients with the anti-α-synuclein approach should begin at very early PD stages or even in a preclinical phase of the disorder. Both of these stages are hard to identify based on contemporary diagnostic methods. The toxic properties of α-synuclein are intensified by cell-to-cell spread and aggregation of endogenous species in newly invaded cells [53].

Recent studies indicated an essential role of low-density lipoprotein receptor-related protein 1 (LRP1) in α-synuclein spreading between cells [54]. LPR1 is a highly expressed central nervous system protein receptor involved in intracellular signaling and endocytosis. It is located predominantly in the plasma membrane. Chen et al. [54], using LRP1 knockout (LRP1-KO) induced pluripotent stem cells (iPSCs), demonstrated that LRP1 is a vital regulator of α-synuclein neuronal uptake and mediator of its spread in the brain. LRP1 binds to the N-terminus of α-synuclein through lysine residues, being a key regulator for the endocytosis of both monomeric and oligomeric forms of α-synuclein. The new findings of LRP1’s role in α-synuclein trafficking point to a potential novel target for synucleinopathies treatment.

Ozdilek and Agirbasli [55] recently demonstrated that the serum LRP1 concentration is associated with the factors influencing the prognosis of PD and disease duration and severity. Moreover, there is a correlation between LRP1 levels and abnormal aggregation of α-synuclein and other IDP between sLRP1 levels and the duration of disease [55].

Iba et al. [56] investigated the impact of aging and inflammation on the pathogenesis of synucleinopathies. They used a mouse model of DLB/PD induced by intrastriatal injection of α-synuclein preformed fibrils (PFF). Aging caused more extensive α-synuclein accumulation in the striatum and amygdala, more significant infiltration of T cells, microgliosis, and behavioral deficits. Transcriptomics analysis revealed that this network’s most important upstream regulators comprised CSF2, LRG, TNFa and poly rl:rC-RNA. These results point to a key role in aging-related inflammation. More specifically, this data shows that CSF2 affects the consequences of α-synuclein spreading and suggests that targeting neuro-immune responses might be a step in developing treatments for DLB/PD. These results are also consistent with the perception that α-synuclein aggregates might lead to neurodegenerative changes by dysregulating adaptive and innate immune responses.

Various factors affect α-synuclein aggregation and fibrillation [26,27,34,44,57,58,59], including the inhibitory effect of β-synuclein on α-synuclein aggregation and toxicity [60,61]. In addition to this, a growing number of recent studies point to the role of defects in proteostasis, autophagy, and lysosomal function in α-synuclein intracellular accumulation and clearance. For example, impairment of lysosomal function due to mutation of PARK9 (a lysosome-related transmembrane P5 ATP transport enzyme, ATP13A2) affects the metabolism of α-synuclein. It causes the development of PD and other synucleinopathies [62].

α-Synuclein forms distinct spatial structures or strains at the very early stages of synucleinopathies, which are potential targets for early diagnosis and treatment. These molecular forms can be detected by various modern techniques, such as seed amplification assays (s) or protein misfolding cyclic amplification (PMCA) [63]. The combination of seed amplification assay with oligomers-specific ELISA (enzyme-linked immunosorbent assay) allows the identification of patients with PD or DLB with high sensitivity and specificity. Moreover, the methods provide vital information about the patient’s clinical disease stage and the severity of the PD clinical symptoms [64]. 

Recent modifications of the PMCA identify the distinct α-synuclein strains specific for different human diseases. They are successfully used for differential diagnosis of patients with PD, multiple system atrophy, DLBs and other disorders using samples of CSF, olfactory mucosa, submandibular gland biopsies, skin, and saliva even at the prodromal stages of the disease [63]. 

Results of a recent study by Emin et al. [65] point to the most toxic types of synuclein aggregates. The authors developed a modification of α-synuclein aggregation reactions in the test tube with subsequent separation of aggregates by size. Wild-type α-synuclein aggregates smaller than 200 nm in length induced inflammation and permeabilization of single-liposome membranes, whereas larger in-size aggregates were less toxic. These initial results were confirmed by characterization of soluble aggregates extracted from post-mortem brains. Aggregates from the brain had a similar size and structure as the small aggregates made in aggregation reactions in vitro. Further experiments demonstrated that the soluble aggregates from PD patients’ brains were smaller and more inflammatory than the large aggregates in control brains. The authors conclude that the small non-fibrillar α-synuclein aggregates are the critical species causing neuroinflammation and disease progression [65].

## 5. Aggregation of β-Synuclein and γ-Synuclein and Their Role in Diseases

α-Synuclein is more aggregate-prone than β-synuclein and γ-synuclein which can be explained by differences in amino acid sequences and the intra-chain dynamics [66]. β-Synuclein is a non-amyloidogenic homolog of α-synuclein and a natural negative regulator of α-synuclein aggregation [60,61,67,68]. Biere et al. [69] show that while α-synuclein forms the fibrils over time, no fibrillation could be detected for β- and γ-synuclein under the same conditions. However, under conditions that significantly speed up aggregation, γ-synuclein can form fibrils with a lag phase approximately three times slower than α-synuclein [69]. Importantly, β-synuclein acts as a molecular chaperone elongating the lag phase of α-synuclein aggregation [70]. β-Synuclein is capable of delaying or preventing α-synuclein fibril formation both in vitro and in vivo.

β-synuclein may be responsible for certain forms of neurodegenerative diseases. For example, the P^123^H and V^70^M mutations in β-synuclein enhanced lysosomal pathology, play a causative role in neurodegeneration and are associated with dementia with Lewy bodies (DLB) [71,72]. In contrast to α-synuclein, two other members of the synuclein family are not found in Lewy pathology [73].

β-Synuclein level is significantly elevated in the CSF of the pre-AD continuum since the preclinical stage. An increase of β-synuclein level in CSF occurs when t-tau and nerve fiber layer (NFL) levels are not yet significantly elevated. Therefore, it represents an emerging biomarker of synaptic damage in this disorder, which may be helpful even at the preclinical stage [19].

Due to the low level of folding, the conformation of synuclein family members is highly dynamic and they can modulate each other’s aggregation propensity. Elevated temperature, low pH, and high concentrations intensify the fibrillation rate of α- and γ-synuclein, while β-synuclein forms fibrils only at low pH. The high molar ratio of β-synuclein inhibits the fibrillation in α- and γ-synuclein, but preformed fibrils of β- and γ-synuclein do not affect fibrillation of α-synuclein [74].

Remarkably, increasing β-synuclein and γ-synuclein affinity to the membrane by point mutations converts these members of the synuclein family into monomers associated with the membrane, increases their cytotoxicity, and predisposes them to form round cytoplasmic inclusions [9].

The presence of γ-synuclein-positive inclusions in motor neurons of amyotrophic lateral sclerosis patients and its ability to aggregate in vitro and in vivo [47,48,75,76] suggest that a term “γ-synucleinopathy” may also be used. 

Williams et al. [77] used a series of chimeras of α-synuclein and β-synuclein to probe the relative input of the N-terminal, C-terminal, and the central NAC domains to the inhibition of α-synuclein aggregation. Measurements of the rates of α-synuclein fibril formation in the presence of the chimeras indicate that the NAC is the primary determinant of self-association leading to fibril formation. However, the N- and C-terminal domains play critical roles in the fibril inhibition process. These findings provide proof that all three domains of β-synuclein together contribute to providing effective inhibition. Knowledge about such multi-site inhibitory interactions spread over the length of synuclein chains may be essential for the development of therapeutics mimicking the inhibitory effects of β-synuclein.

Interestingly, α-synuclein fibrils made in the presence of β-synuclein are less cytotoxic and possess decreased cell seeding capacity. Moreover, they are more resistant to fibril shedding compared to α-synuclein fibrils alone [23,78]. NMR studies showed that the overall structure of the core of α-synuclein when co-incubated with β-synuclein is minimally perturbed. However, the dynamics of Lys and Thr residues in the imperfect KTKEGV repeats of the α-synuclein N-terminus are increased. Thus, amyloid fibril dynamics play a crucial role in modulating synuclein toxicity and seeding. 

Interestingly, β-synuclein is able to potentiate synaptic vesicle dopamine uptake and rescue dopaminergic neurons from MPTP-toxic effect in the absence of other synucleins [23]. Potentiation of dopamine uptake by β-synuclein may be explained by different protein architecture of the synaptic vesicles. These results demonstrate that of the three members of the synuclein family, only β-synuclein can potentiate VMAT-2–dependent uptake of dopamine and structurally similar molecules by synaptic vesicles. The authors hypothesize that the elevated level of β-synuclein-triggered complexes at the synaptic vesicles, and not the absence of other synucleins *per se*, causes the reduced sensitivity to MPTP toxicity in mice lacking α-synuclein and/or γ-synuclein. These results point to a critical role of β-synuclein in the resistance to MPTP toxicity by dopaminergic neurons lacking the usual balance of this protein and other members of the synuclein family [23].

## 6. Inhibitors of α-Synuclein Aggregation and Fibrillation as Potential Tools for Therapy

α-Synuclein has been identified as a key target for the development of therapeutic approaches to synucleinopathies given its central role in the pathophysiology of these diseases. Treatment strategies can be classified into several groups [79]: (1) inhibition of formation of toxic α-synuclein aggregates (aggregation inhibitors), (2) lowering α-synuclein expression (e.g., antisense therapy), (3) removal of toxic α-synuclein aggregates (immunotherapy), (4) removal of toxic forms of α-synuclein by enhancement of cellular clearance mechanisms (autophagy, lysosomal microphagy), (5) modulation of neuroinflammatory processes and (6) neuroprotective strategies. Current therapeutic approaches and elucidation of promising future treatment approaches are described in a recent article [79].

The results of recent experiments suggest that therapeutic downregulation of α-synuclein expression in PD patients may be a mostly harmless choice since it should not lead to adverse side effects on the functions of the nigrostriatal system [22].

High-throughput screening assays are efficient methods usually used to analyze a huge number of potential inhibitors of α-synuclein aggregation and to identify among them the most potent compounds. Kurnik et al. [80] used a combination of SDS-stimulated α-synuclein aggregation with Förster resonance energy transfer (FRET) to characterize the initial stages of α-synuclein aggregation. After screening 746.000 compounds, the authors identified 58 hits that markedly inhibit α-synuclein aggregation and reduce membrane permeabilization activity. The most effective aggregation inhibitors are derivatives of (4-hydroxy-naphthalen-1-yl) sulfonamide. They interact with the N-terminal part of monomeric α-synuclein with high affinity and reduce early-stage oligomer-membrane interactions. Notably, some of them also reduce α-synuclein toxicity toward neuronal cell lines.

Several approaches have been used to prevent or diminish the toxic effects of synuclein aggregation and fibrillation. One of them includes reducing α-synuclein expression, for example, by antisense oligonucleotides or nucleic acids. Other methods are based on the inhibition of toxic α-synuclein aggregates build-up using aggregation inhibitors. Some strategies are directed to dissolve and eliminate the existing intracellular or extracellular toxic α-aggregates, for instance, by immunotherapy. Various forms of α-synuclein induce an immune response including inflammation, highlighting the immune function of α-synuclein. Aggregation may render a protein “foreign” to the immune system and cause immunoresponce [81].

To reduce the toxic effect of α-synuclein, methods based on the enhancement of cellular clearance mechanisms removing toxic α-synuclein aggregates are used more often. Some of them are directed to the activation of autophagy or lysosomal microphagy. 

An example of an approach based on the activation of autophagy gives hope for future therapeutic applications [82]. For instance, a small molecular inhibitor for prolyl oligopeptidase, KYP-2047, relieves α-synuclein-induced toxicity in various PD models by inducing autophagy and preventing α-synuclein aggregation, without reducing soluble α-synuclein oligomers. The authors hypothesize that KYP-2047 decreases the concentration of those toxic forms of α-synuclein that were not detected by ELISA assay specific for soluble α-synuclein. These findings indicate that when considering potential drug treatment for synucleinopathies, it is critical to characterize the α-synuclein aggregation process in detail, identifying the most toxic forms of α-synuclein. The authors also assume that previous failure to use α-synuclein aggregation inhibitors may be explained by the fact that removal of random α-synuclein aggregates forms does not provide neuroprotection [82].

Other methods, including modulation of neuroinflammatory processes and neuroprotective strategies, are currently under development. Since there is currently no method that guarantees the complete elimination of α-synuclein toxicity, new approaches and novel substances discussed below are currently under investigation.

A promising methodology for the development of new amyloid inhibitors is recently described in Murray et al. article [44]. The authors used a de novo structure-based computational design to create 35–50 residue amyloid capping inhibitors that bind to the growing ends of amyloid fibrils. The principle of the method is not restricted to α-synuclein, but can be applied to tau, beta-amyloid fibrils, and other fibril-forming proteins. Their structure was determined using Cryo-EM, ssNMR spectroscopy, and MicroED. The designed inhibitors can bind to the ends of fibrils, “capping” them and thus preventing further growth, seeding, and reducing the toxicity of such proteins both in vitro and in vivo. Since such inhibitors specifically bind to an end of an amyloid fibril, averting its elongation, other end of the fibril might be able of growing. However, analysis of this growth has shown that fibrils grow mainly unidirectionally, avoiding this issue. Importantly, inhibitors that decrease α-synuclein primary aggregation are also effective at decreasing cellular seeding. 

Another promising approach to finding specific inhibitors of α-synuclein toxic species is the use of aptamers binding with high specificity to different truncated forms of α-synuclein fibrils with no cross-reactivity toward other amyloid fibrils [83]. Testing a panel of aptamers allowed to identify two of them (Apt11 and Apt15) possessing higher affinity to most C-terminally truncated forms of α-synuclein fibrils. Apt11 inhibited α-synuclein-seeded aggregation in vitro. Apt11 decreased the insoluble phosphorylated form of α-synuclein at Ser^129^ (pS^129^-α-syn) in a cell model. This aptamer inhibited α-synuclein aggregation assessed by RT-QuIC reactions seeded with brain homogenates isolated from PD patients. These aptamers may be used as tools for research and diagnostics or therapy.

Natural compounds also possess antiaggregational potential. Many phytochemicals inhibit α-synuclein aggregation and possess a neuroprotective effect [84,85,86]. For example, a polyphenolic component derived from green tea—epigallocatechin 3-gallate (EGCG) binds to α-synuclein by hydrophobic interactions and suppresses its aggregation at 100 nM in a concentration-dependent manner [85]. 

Curcumin is a biphenolic phytochemical compound from the root of *the Curcuma longa* efficiently inhibits α-synuclein from turning into an amyloid [87]. Curcumin prevents aggregation of both wild-type and mutant forms (E^46^K and H^50^Q) of α-synuclein (Figure 1). Moreover, curcumin destabilizes preformed α-synuclein amyloid aggregates and regulates α-synuclein amyloid formation [86]. Interestingly, that curcumin induces hypermethylation of γ-synuclein gene through the upregulation of the DNA methyltransferase 3 (DNMT3) which reduces γ-synuclein expression [87]. These results show that polyphenol compounds may have therapeutic implications in PD treatment [88].

In addition to many inhibitors of synuclein aggregation, several substances enhancing α-synuclein aggregation (proaggregators) have been identified. For example, several phenyl-benzoxazol compounds, e.g., 2-arylbenzoxazole (PA86, 1) enhance the α-synuclein aggregation [80]. In another study TKD150 and TKD152 were investigated as proaggregators for α-synuclein. In comparison to a previously reported proaggregator, PA86, these two identified compounds were able to promote aggregation of α-synuclein at twice the rate [58]. These studies are important to better understand the intimate mechanism of α-synuclein aggregation.

## 7. Conclusions and Future Directions

There are several directions in synucleins studies which definitely will bring better understanding of their normal functions, association with diseases and open new strategies for the early diagnosis and timely treatment of synucleinopathies. Although the majority of investigations are still directed to α-synuclein, the functional role of two other members of the family is becoming clearer. The existing data from the literature on synucleins is descriptive, and a transition from observational to interventional study designs is currently in need. It is evident that this conversion may be slow, but hopefully it would lead to a breakthrough due to the development of multiomics and other novel research approaches. 

Currently, many studies are directed to the understanding of the role of α-synuclein in the synaptic region and dopamine vesicle regulation. At the same time, its interaction with other cellular components remains poorly understood and many intriguing questions concerning synuclein functions remain unclear. For example, does α-synuclein interact with mitochondrial DNA directly and influence transcription profile? What role synucleins play in endoplasmic reticulum-Golgi traffic [89]? Recent studies investigating microtubule dynamics showed that α-synuclein facilitated the formation of short microtubules and favorably binds to 4 protofilaments. However, synuclein’s role in the axonal transport and interaction with microtubules remains to be determined [90].

One of the critical directions in future work is the development of α-synuclein based biomarkers. Timely identification of PD symptoms should allow the beginning of early treatment. 

### 7.1. Importance of Easily Accessible Samples for Diagnosis

Relevant biomarkers for α-synuclein pathology in PD are emerging, as well as blood-based markers of general neurodegeneration and glial activation [91,92]. Detection of misfolded α-synuclein in easily accessible fluids, e.g., blood, CSF, and skin would be a great step forward for early diagnosis of PD and other synucleinopathies [91,92,93]. Preliminary results indicate that skin biopsies, including nerve terminals, can be used in seeding aggregation assays to detect misfolded α-synuclein in PD, DLB, and multiple system atrophy [94]. Other potential peripheral tissues for detecting misfolded α-synuclein include the olfactory mucosa and the submandibular gland [63,94]. They can be used for both early diagnosis and also for the analysis of medication’s efficiency. Biomarkers that can measure relevant treatment effects downstream of the drug target will likely be used more frequently to optimize go/no-go decisions on moving drugs forward. In the future, such biomarkers will be used in clinical routine practice to monitor the efficacy of disease-modifying therapies in individual patients.

### 7.2. Role of the Gut-Brain Axis

Recent findings point to a highly complex relationship between the gut and the brain in PD, providing the potential for the development of new biomarkers and therapeutics. The neuropathological process that leads to PD seems to start in the enteric nervous system or the olfactory bulb and spreads via rostro-cranial transmission to the substantia nigra and further into the CNS. This raises the exciting possibility that environmental substances can trigger pathogenesis [94,95]. The discovery of importance of the gut-brain axis in PD development opens a new approach for the search of therapeutic targets and biomarkers detection [95,96]. However, there are definite gaps in our knowledge and new research is needed for a better understanding of the role and interrelationships of α-synuclein with the vagus nerve, with the enteric nervous system, altered intestinal permeability, and inflammation [96,97].

Reduced activity of protein degradation systems, such as proteasomes, autophagic systems and lysosomes occur in patients with synucleinopathies. Therefore, the activation of autophagy gives hope for future therapeutic applications [36,79,82,83]. There are approaches that are not aimed at α-synuclein in a straight line but affect it indirectly. Autophagy is critically related to the anomalous accumulation of α-synuclein, and defects in lysosome-related transmembrane ATP transport enzyme ATP13A2 (also known as PARK9) impair lysosomal function and inhibit the toxicity of α-synuclein. Therefore, this enzyme may be considered a possible target in synucleinopathies [36,62]. Other autophagy activation methods also give hope for future therapeutic applications [82]. 

One of the promising agents causing the reduction of α-synuclein oligomer accumulation is GNE-7915, specific inhibitor of the long-term brain-penetrant leucine-rich repeat kinase 2 (LRRK2 or PARK8). This inhibitor significantly decreases oligomers in the striatum without causing adverse peripheral effects [97]. Inhibition of mutant LRRK2 hyper-kinase activity to normal physiological levels offers an encouraging and safe disease-modifying therapy to improve the course of synucleinopathies. 

Small molecular inhibitors for prolyl oligopeptidase (PREP), for example, KYP-2047, relieve α-synuclein-induced toxicity, enhancing autophagy and preventing α-synuclein aggregation [83].

KYP-2047 had a significant positive impact on total oligomeric α-synuclein and particularly on proteinase K resistant α-synuclein species. There is hope that further modifications of the method will allow KYP-2047 and similar inhibitors to improve brain penetration based on the new delivery strategies. These strategies may include the fusion of effective inhibitors with a cell-penetrating peptide-based tag or conjugation to other types of delivery constructs such as brain-penetrating nanoparticles. The inhibitor designs are also genetically encodable, including viral delivery, another possible transport mechanism [82].

There is hope that scientific progress in this field will translate to the successful use of new diagnostic, prognostic, and therapeutic approaches that will ultimately improve patients’ lives.

## Figures and Tables

**Figure 1 biomedicines-10-03241-f001:**
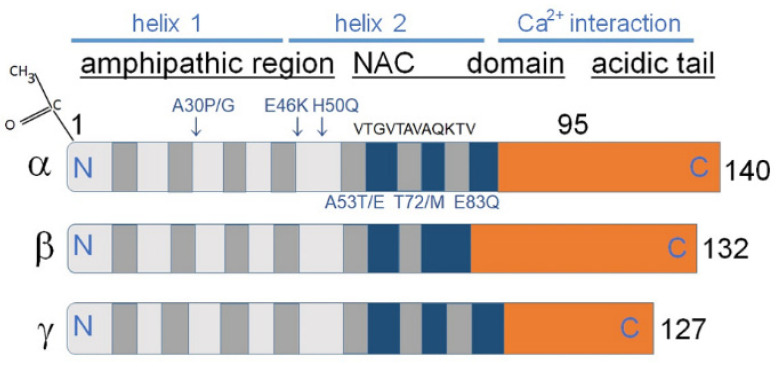
The core sections of the seven amino-terminal repeats (KTKEGV motive) are shown as dark grey bars in α and γ-synucleins. These motives are located between amino acids 7–87 of human α- and γ-synuclein. β-Synuclein has only six such repeats. Positively charged regions are shown in light gray, hydrophobic regions are blue, and negatively charged regions are light brown. In α-synuclein the non-amyloid-β component (NAC) is shown, containing amino acids 61–95, playing a key role in aggregation in vitro. The absence of this region in β-synuclein decreases its tendency to aggregate. Amino acids 71VTGVTAVAQKTV82 are the most essential for α-synuclein aggregation [8,9]; it is. This domain is partly absent in β-synuclein and to some extent conserved in γ-synucleins, which might clarify why both homologs of α-synuclein are not aggregation prone as α-synuclein. Some missense mutations predisposing to diseases (A^53^E, A^53^T, A^30^P, E^46^K, H^50^Q, T^72^M, and E^83^Q) are displayed as blue letters above and under the α-synuclein image (Modified from [10]). At the upper left corner N-terminal acetylation of α-synuclein is shown, which slows down its aggregation and changes the morphology of the aggregates [11,12].

**Figure 2 biomedicines-10-03241-f002:**
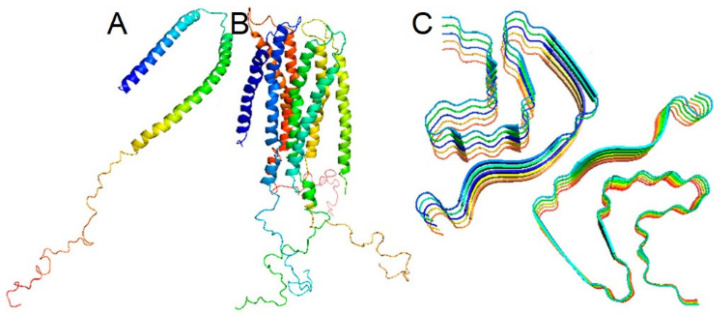
Micelle-bound, tetrameric, and fibrillar α-synuclein. (**A**)—α-Synuclein is disordered under physiological conditions in solution. Disordered form is in equilibrium with a minor α-helical tetrameric form in the cytoplasm (**B**) α-Synuclein is α-helical when bound to a cell membrane (**C**) α-Synuclein may form polymorphic amyloid fibrils with unique arrangements of cross-β-sheet motifs. From Korneev et al. [30].

## Data Availability

Not applicable.

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
