# Peer review of "Synucleins: New Data on Misfolding, Aggregation and Role in Diseases"

_biomedicines, 2022, doi:10.3390/biomedicines10123241_

Round 1
Reviewer 1 Report
The manuscript presented by Dr. A. Surguchov is a review summarizing the latest data on synuclein family proteins and their role in neuropathology. The review focuses more on synuclein-based methods of diagnosis and perspectives of alpha-synuclein as a target in the treatment of Parkinson's disease and other synucleinopathies, but it also sufficiently covers issues of synucleins gene organization and structural features of translation products. At the same time, much less attention has been given to the review physiological functions of synucleins. The author draws a number of important conclusions that can serve as a starting point for future research. Overall, the manuscript has a clear and distinct structure and may be of interest to a wide range of researchers. At the same time, the authors are encouraged to address one minor issue before the manuscript can be recommended for acceptance.
Minor claim
Line 104. The cellular effects of synuclein gene inactivation would be extremely useful to complement the data on the multidirectional effects of synucleins, especially gamma-synuclein, on CNS function.
Author Response
Reviewer 1: The physiological role of synucleins should be discussed in more detail.
We thank the reviewer 1 for this suggestions. The physiological role of synucleins are discussed in sections 2.2 and 2.3.
Reviewer 1. The cellular effects of synuclein gene inactivation would be extremely useful to complement the data on the multidirectional effects of synucleins, especially gamma-synuclein, on CNS function.
In response to reviewer's comment about the multidirectional effects of synucleins, especially gamma-synuclein, on CNS function we added the following text and reference after lines 104-107 ending by "..synaptobrevin-2 [14}"
”Experiments with g-synuclein homozygous knockout mice demonstrated that inactivation of the gene caused the improvement of working memory. On the other hand, behavioral tests assessing spatial learning and memory, i.e. Morris water maze and Object location tests, did not reveal changes between g-synuclein knockouts and control mice. These results demonstrate that g-synuclein is directly involved in the regulation of cognitive functions” [18]
Kokhan et al., Differential involvement of the gamma-synuclein in cognitive abilities on the model of knockout mice. BMC Neurosci. 2013 May 14; 14:53.
Reviewer 2 Report
- Authors should describe the physiological functions of synucleins in more detail (Par 2.2);
- Authors should outline informations of Paragraph 2.2. relative to the role of synucleins in pathology in a table, in which they report the pathology, the levels of synucleins and the associated references;
- Authors should describe in more detail the aggregation process of alpha-synuclein, also emphasizing the role of oligomeric conformers, that are actually suggested as the most toxic species able to prompt neuronal dysfunction and cell death. A scheme representing the aggregation process would be very useful.
- Lines 200-208 are not included in the correct paragraph, that is entitled "Post-translational modifications (PTMs) of alpha–synuclein". Collectively, I believe that Par. 3.1 can be directly incorporated in Par. 3.0;
- Authors should expand this point (lines 239-249): "Alternative tactics to affect aggregated α-synuclein may be also beneficial."
- Actual Par. 3.3. should be moved after 3.1 (so becoming the new 3.2). At the same time, Paragraph regarding synuclein-based diagnostic approaches should precede the one on possible therapeutic strategies (included in Par. 3.2 and 6).
- Par. 5 on the aggregation of beta and gamma synuclein should be put after the one concerning alpha-synuclein aggregation.
- Paragraph 4 is not completely focused on diagnostic strategies for synucleinopathies (see lines 338-345). This paragraph should be focused on the detection of synuclein in body fluids, peripheral tissues and molecular imaging. Not completely inherent information should be included in other paragraphs. Paragraph 4 should also include information on the diagnostic relevance of beta-synuclein levels (lines 397-391).
In summary, I suggest a reorganization of the review as follows:
- Introduction;
-Stucture, function and role in pathology;
- Aggregation process of all the three proteins;
- Diagnostic strategies;
- Possible therapeutic approaches.
Author Response
- Authors should describe the physiological functions of synucleins in more detail (Par 2.2).
- Thank you for this suggestion. The physiological functions of synucleins are described in Section 2.2
- Authors should outline informations of Paragraph 2.2. relative to the role of synucleins in pathology in a table.
- Thank you. The information about the role of synucleins in pathology is very controversial, and attempts to combine them in a Table may be confusing. We made this conclusion after analyzing the following publications:
- Malec: Testing ASN in plasma is potential test for diagnose PD, but previous studies are controversial. Plasma ASN level is not valuable marker of the disease. It does not differ in subtypes of the disease.
- Chun-Wei Chang et al. :”controversial results regarding the correlation between motor severity and α-synuclein levels in peripheral blood from patients with PD”
- Albillos et al., Studies have revealed controversial results regarding the diagnostic accuracy of plasma α-synuclein levels in patients with Parkinson's disease (PD)
- Mavroudis I et al., Alpha-synuclein Levels in the Differential Diagnosis of Lewy Bodies Dementia and Other Neurodegenerative Disorders: A Meta-analysis. Alzheimer Dis Assoc Disord. 2020 Jul-Sep;34(3):220-224. doi: 10.1097/WAD.0000000000000381.
- Malec-Litwinowicz M et al., The relation between plasma α-synuclein level and clinical symptoms or signs of Parkinson's disease. Neurol Neurochir Pol. 2018 Mar;52(2):243-251. doi: 10.1016/j.pjnns.2017.11.009.
- Authors should describe in more detail the aggregation process of alpha-synuclein.
- The aggregation of synucleins is described in Section 3.
Lines 200-208 are not included in the correct paragraph, that is entitled "Post-translational modifications (PTMs) of alpha–synuclein". Collectively, I believe that Par. 3.1 can be directly incorporated in Par. 3.0;
-In order to make these parts of the texts more compatible and matching each other we added to the title of Section 3.1 the following :"and other factors affecting aggregation"
Thank you very much for your thoughtful suggestions
Reviewer 3 Report
1. Authors discussion about domain identity in lines 80-87, can be easily depicted by a diagram or protein structural modification cartoons drawn by using any bioinformatic or computational tool, therefore authors are advised to include that in order to make their review more interactive.
2. In lines 105-106 authors are asked to be more specific and rephrase these lines accordingly, the emphasis on dopamine transmission after mentioning about regulation of more generic function of vesicle pool regulation is unnecessary. As of now the available data indicates that α-Synuclein mediated regulation is not limited only to the dopamine-carrying vesicles. Therefore, authors are advised to consider rephrasing of these lines.
3. Lines 109…. Authors are advised to look for and include (if possible) the literature on α-Synuclein chaperoning function via SNARE complexes other than VAMP2, those are involved in trafficking other than the transport of neurotransmitters. Synaptic defects are one aspect of neurodegeneration and is apparent relatively at the late stage of the disease, therefore other SNARing components can not be overlooked. Please revise.
4. Lines 143, the conclusion drawn about the fluctuation of Synuclein expression levels in response to MPTP, authors are advised to add in/describe the information on the domains which were tested to quantified the expression levels of various synucleins in the cited literature.
5. Lines 158-159 authors are asked to explain what they mean by the “rigorousness of the disorder”, authors themselves talked about Autism-spectrum in earlier lines?
6. Lines 190-191, is toxicity a gain of function or is it the gain-of-function that is responsible for the toxicity? Can authors elaborate on this as per their find or understanding from the recent enhancements in the field?
7. Line 197, the conclusion made is too early becasue neither the α-Synuclein interaction and involvement with other SNARE proteins is fully described nor the Synaptic physiology is fully understood. Authors are advised to rephrase or remove this conclusion accordingly.
8. Section 3.2, author’s explanation on failed clinical trials lacks the consideration of the fact that PD is an entire bundle of disease spectrum, therefore any PD oriented therapy targeted to signal molecule is likely to have such outcomes, therefore authors are advised to include this perspective to their explanations.
9. Lines 266-67, when author’s talk about the origin of patient homogenate what they are actually referring to in terms of patient characterizations. Are they referring to “UPDRS Score, ROME-III Questionnaire, RBDSQ Score, Hoehn & Yahr Score, MOCA Score, Sniffin' Stick Score etc.” or some other characterization/specification, please clarify and modify the text accordingly?
10. Line 382, were their distinct morphological changes among the brain regions?
11. Lines 441-444, does β-synuclein mediated potentiation is specific to dopamine carrying vesicles? What are author’s views in context to state-of-art.? Please modify the manuscript accordingly.
12. Section-6, there are some recent works, where anti-bodies against α-Synuclein are proposed as a therapeutic possibility, why authors didn’t consider this aspect in their review or authors are of a view that such a possibility is a non-significant development in the field?
13. Section-7.2, Along the gut-brain axis authors missed the direct observations from PD patients reported in (10.1016/j.parkreldis.2021.05.022) which are covering the miRNA aspect of the disease.
14. To diagnosis as a challenge in Synucleopathies, a potential recent solution is offered in terms of peptide-printing techniques. Authors may consider including the relevant literature in their review.
15. In section-6, authors endorse and emphasize only α-Synuclein oriented therapies, however in their review for synuclein mediated toxicity, most of their arguments are developing from the findings reported in MPTP based PD-model. It has been reported that the MPTP primate model do not show any Lewy-body formation, to which α-Synuclein is reported as the main component. Therefore, authors are asked to realize such gaps in their manuscript and adapt the changes accordingly.
16. Figures do not appear standardized and picture quality is not good, please work on them.
17. Please revisit the manuscript for other format related, grammatical or typos related issues.
Author Response
- Reviewer 3. Authors discussion about domain identity in lines 80-87, can be easily depicted by a diagram or protein structural modification cartoons drawn by using any bioinformatic or computational tool, therefore authors are advised to include that in order to make their review more interactive.
- Thank you. We believe that such diagrams and cartoons can be easily found in other reviews on synucleins. We doubt that we can make it in a different way than in other articles.
- In lines 105-106 authors are asked to be more specific and rephrase these lines accordingly
- Thank you, we did it as recommended.
- Lines 109…. Authors are advised to look for and include (if possible) the literature on α-Synuclein chaperoning function via SNARE complexes ...
- Thank you, this information is included in Section 2.2
- Lines 143, the conclusion drawn about the fluctuation of Synuclein's expression levels in response to MPTP, authors are advised to add in/describe the information on the domains which were tested to quantified the expression levels of various synucleins in the cited literature.
- Thank you for this suggestion.
- Lines 158-159 authors are asked to explain what they mean by the “rigorousness of the disorder”, authors themselves talked about Autism-spectrum in earlier lines?
- Thank you. We replaced "rigorousness" on the "severity"
- Section 3.2, author’s explanation on failed clinical trials lacks the consideration of the fact that PD is an entire bundle of disease spectrum, therefore any PD oriented therapy targeted to signal molecule is likely to have such outcomes, therefore authors are advised to include this perspective to their explanations.
- Than you; as recommended, we added:" Finally, PD is heterogeneous disease, including a bundle of disease spectrum, making the treatment difficult."
- Lines 266-67, when author’s talk about the origin of patient homogenate what they are actually referring to in terms of patient characterizations. Are they referring to “UPDRS Score, ROME-III Questionnaire, RBDSQ Score, Hoehn & Yahr Score, MOCA Score, Sniffin' Stick Score etc.” or some other characterization/specification, please clarify and modify the text accordingly?
- We believe that this information is excessive, since the review concerns basically the data about biochemistry, genetics and molecular biology.
- Figures do not appear standardized and picture quality is not good, please work on them.
- Thank you, we did it
- Please revisit the manuscript for other format related, grammatical or typos related issues.
- We appreciate your precious comments and suggestion.
Reviewer 4 Report
This review summarized the most significant recent findings of the role in pathology of three members of the synuclein family . Here they summarized the cellular functions and pathological function of not only the most commonly studied α-synuclein, but also β- and γ-synuclein. That is very interesting and meaningful. This review is well summarized and showed.
Here are some suggestions for improving:
1. Paragraph “3.2. Approaches to reduce α-synuclein toxicity” this topic can be moved back to “6. Inhibitors of α-synuclein aggregation and fibrillation as potential tools for therapy”, and combined them to a new topic : potential therapeutic tools for synuclein related neurodegenerative disease. Part1. 3.2. antibody Approaches to reduce α-synuclein toxicity ; part2. Inhibitors of α-synuclein aggregation and fibrillation as potential tools for therapy.
2. “4. Synuclein-based methods of disease diagnosis.” This topic may be moved after “5. Aggregation of β-synuclein and γ-synuclein and their role in diseases”. Thus, the description order will become more reasonable: pathological role, diagnosis, then therapy.
3. 421-431. This paragraph is largely repeated with 131-148. And didn’t have much connection with aggregation and role in disease. May just simply delete this part.
Author Response
- Reviewer 4. Paragraph “3.2. Approaches to reduce α-synuclein toxicity” this topic can be moved back to “6. Inhibitors of α-synuclein aggregation and fibrillation as potential tools for therapy”, and combined them to a new topic : potential therapeutic tools for synuclein related neurodegenerative disease. Part1. 3.2. antibody Approaches to reduce α-synuclein toxicity ; part2. Inhibitors of α-synuclein aggregation and fibrillation as potential tools for therapy.
- Thank you. We believe, that the suggestion to move the topic to 6. Inhibitors contradicts the suggestions of other reviewers.
- 4. Synuclein-based methods of disease diagnosis.” This topic may be moved after “5. Aggregation of β-synuclein and γ-synuclein and their role in diseases”. Thus, the description order will become more reasonable: pathological role, diagnosis, then therapy.
- Thank you. The suggestion to move this topic contradicts the suggestions of other reviewers.
- 421-431. This paragraph is largely repeated with 131-148. And didn’t have much connection with aggregation and role in disease. May just simply delete this part.
- Thank you, we deleted it.